# Density Functional Theory, Chemical Reactivity, Pharmacological Potential and Molecular Docking of Dihydrothiouracil-Indenopyridopyrimidines with Human-DNA Topoisomerase II

**DOI:** 10.3390/ijms21041253

**Published:** 2020-02-13

**Authors:** Mohamed E. Elshakre, Mahmoud A. Noamaan, Hussein Moustafa, Haider Butt

**Affiliations:** 1Chemistry Department, College of Science, Cairo University, Cairo 12613, Egypt; husseinmam@hotmail.com; 2Department of Mechanical Engineering, Khalifa University, Abu Dhabi 127788, UAE; haider.butt@ku.ac.ae

**Keywords:** thiouracildihydroindenopyridopyrimidine (TUDHIPP), DFT, molecular docking simulation, drug likeness screening, structure activity and property relationships (SAR/SPR)

## Abstract

In this work, three computational methods (Hatree-Fock (HF), Møller–Plesset 2 (MP2), and Density Functional Theory (DFT)) using a variety of basis sets are used to determine the atomic and molecular properties of dihydrothiouracil-based indenopyridopyrimidine (TUDHIPP) derivatives. Reactivity descriptors of this system, including chemical potential (µ), chemical hardness (η), electrophilicity (ω), condensed Fukui function and dual descriptors are calculated at B3LYP/6-311++ G (d,p) to identify reactivity changes of these molecules in both gas and aqueous phases. We determined the molecular electrostatic surface potential (MESP) to determine the most active site in these molecules. Molecular docking study of TUDHIPP with topoisomerase II α and β is performed, predicting binding sites and binding energies with amino acids of both proteins. Docking studies of TUDHIPP versus etoposide suggest their potential as antitumor candidates. We have applied Lipinski, Veber’s rules and analysis of the Golden triangle and structure activity/property relationship for a series of TUDHIPP derivatives indicate that the proposed compounds exhibit good oral bioavailability. The comparison of the drug likeness descriptors of TUDHIPP with those of etoposide, which is known to be an antitumor drug, indicates that TUDHIPP can be considered as an antitumor drug. The overall study indicates that TUDHIPP has comparable and even better descriptors than etoposide proposing that it can be as effective antitumor drug, especially 2H, 6H and 7H compounds.

## 1. Introduction

Computational medicinal chemistry has become an important element of modern drug research. Molecular modeling [1] is used in many fields, particularly drug design [2,3]. Drug design is a science that focuses on the discovery and design of new therapeutic chemicals or biochemical and their development into useful medicines [4]. There has been a growing interest in the study of the chemical and biological aspects of heterocyclic compounds in the medicinal chemistry field in the past decades.

Organic compounds with heteroatoms such as N, O and S have attracted great attention due to their wide applications in medicinal chemistry research. Heterocyclic compounds are cyclic organic compounds with at least two different elements, including nitrogen, oxygen and sulfur, as ring member atoms [1]. Among the major pharmaceutical natural products and synthetic drugs, heterocyclic compounds [4,5,6,7,8,9] have a distinct place, because of their remarkable biological activities and established application in medicinal chemistry [5]. The importance of the sulfur atom in drugs in the form of sulfide or disulfide linkages provides great stability for the three dimensional structure of the molecule [10].

We are proposing the investigation of the biological and pharmacological activities of the novel thiouracilindenopyridopyrimidines (TUIPPs), based on the proven outstanding biological activity the analogous indenopyridopyrimidine (UIPP) derivatives [11], which have oxygen and nitrogen atoms as heteroatoms. TUIPPs are expected to exhibit even higher chemical, biological, and pharmacological activities than UIPPs because of the extra sulfur atom in the former [10]. A unique feature of both TUIPPs and UIPPs is their ability to exist in the dihydro and oxidized forms in the cell tissues, as shown in Figure 1 for compound 1, which shows the oxidation of compound 1H to compound 1O. We focus on the nine derivatives of the TUIPP dihydro form 1H–9H, as shown in Figure 2, which are named from now on as thiouracil-based dihydroindenopyridopyrimidines (TUDHIPPs), which have been synthesized and characterized by Hassaneen et al. [12]. It is well established that compounds 1H–9H are prodrugs and 1O–9O are drugs [11]. Our main objective is to explore the electronic structure properties and the pharmacological activity of the nine derivatives of TUDHIPP using DFT and molecular docking modeling.

There is a global concern about the widespread of cancer, which is considered as one of the major causes of mortality worldwide [13]. Chemotherapy is one of the strategies in cancer treatment, where one or several drugs can be used in cancer treatment. However, these drugs cannot play their role because detoxification enzymes attack these drugs. Both the dihydro and oxidized forms of the uracil-based indenopyridopyrimidines (UDHIPPs) and UIPPs, prove to be very powerful anticancer drug for treatment of human cervical cancer, human T-cell leukemia, human mammary carcinoma, human NSCLC, human colon cancer, human glioblastoma, melanoma, and human prostate cancer [11]. Therefore we expect that TUDHIPPs, which have an extra sulfur atom, to be more powerful and more effective drugs for cancer treatment.

The process of drug development is not straightforward, time-consuming, and cost intensive. It requires several years for lead identification, optimization [14], and in vitro and in vivo [15,16,17,18,19,20] testing before the first clinical trials [21,22]. During the process of drug discovery, a huge set of complex data with remarkable uncertainty levels is produced [23]. In order to efficiently utilize the data to quickly target compounds with a good balance of properties [24] and in order to overcome the data uncertainty, multi-parameter optimization (MPO) methods are proposed. 

Density Functional Theory (DFT) can allow us to identify and derive many useful and important concepts in chemistry [8,25,26]. To study the proposed compounds, we use the DFT approach, which is a very useful and convenient framework of the discussion of chemical reactivity [27]. Among various computational methodologies, DFT has become more popular for calculating molecular properties, which has been extensively used to compute the electronic structure properties in the ground and excited electronic states of molecules in both gas and aqueous phases [28].

Molecular docking studies are very useful computational and simulation techniques that allow the assignment of the pocket in which the drug candidates can perfectly fit within the protein. Molecular docking studies can also allow the assignment of binding sites and calculating binding energies between drug candidates and proteins, giving insights of how the drug works inside the cell [29,30]. 

Quantitative Structure Activity Relationships (QSARs) [8,25,31,32,33,34,35,36] attempt to establish a correlation between the physicochemical parameters of chemical structures and their biological activity. There is a general principle of medicinal chemistry on which QSAR is based which connects the biological activity of a compound to its molecular structure, which states that structurally similar molecules may have similar biological activities [37]. The information on the structure of molecules is encoded in molecular descriptors and the various models of QSAR define mathematical relationships between biological activities of known ligands and descriptors are used to predict biological activities of unknown ligands. The methods of QSAR have been used to predict and classify biological activities [38,39] of virtual or newly synthesized compounds in chemistry, biology, toxicology and drug discovery [40,41,42,43,44]. In addition, models of QSAR can be used in the design of new chemical compounds. QSAR models are now considered as essential and efficient tools in pharmaceutical industries in the early stages of drug discovery [23,43,44,45].

There are parameters of critical importance in the development of new drug molecules due to their role in defining overall safety margins, dose intervals, and dose amounts [46], which include Absorption-Distribution-Metabolism-Excretion(ADME). The meticulous investigation of physico- chemical parameters is essential since medicinal chemists can relate quite easily to such physicochemical parameters [47]. Designing a new drug needs to find the right balance of physicochemical properties and ADME parameters. Therefore, it is crucial to test if TUDHIPP derivatives satisfy the ADME prerequisites as anticancer drugs. 

Hence, it is important to study molecular geometry, electronic structure properties and substituent effect of TUDHIPP using ab initio and DFT methods in order to explore their chemical reactivities. It is also essential to study molecular docking of these molecules in order to explore the binding sites and binding energy with human topoisomerase *II α* and *β*. Thirdly, to study their pharmacological relevance as potential drugs using Lipinski and Veber’s rules. Finally, to establish a correlation between their physicochemical properties and ADME parameters to find out their appropriateness as potential anticancer drugs.

This is a novel class of molecules that has not been investigated before. The proposed molecules can be potential anticancer drugs by comparison with etoposide which is a well- established anticancer drug.

## 2. Results and Discussion

### 2.1. Methods Benchmark

The choice of computational method and basis set is quite crucial to the accuracy of calculations. Geometry optimization of the thiouracil (TU) nucleus (Figure 3a) was achieved using three different levels of theory. The non-electron *correlated ab initio/HF,* the electron correlated ab initio/MP2, where both apply the wavefunction Schrodinger equation, and Density Function Theory DFT, which applies electron density approach using two different basis sets, namely *6-311++G* (*d, p*), and *cc*-*pVDZ*. The bond lengths from geometry optimization results are listed in Table 1, which are compared with the bond lengths results from x-ray diffraction [48,49,50], to determine the optimum computational approach which reproduces the experimental x-ray geometrical parameters. A comparison of bond lengths calculated with various methods using the 6-311++G (*d*, *p*), and cc-pVDZ basis set of TU is shown in Table 1. These results show that DFT/B3LYP method gives better agreement with experimental data [48,49,50] than the ab initio HF and MP2 methods, as indicated by the values of the mean absolute deviation A%, which is calculated as the mean absolute deviations between the calculated and experimental values for each method, as given in the last row of Table 1. These results indicate that the DFT/B3LYP/6-311++G (d, p) method is the most suitable for predicting the bond length of TU. Therefore it is adopted for the rest of the calculations.

### 2.2. Comparison of Atomic Charges Methods

The calculation of atomic charges using computational quantum chemistry in any molecule plays a central role in the determination of its reactivity. There are three methods by which atomic charges are calculated in molecular modeling, namely: the Mulliken, ChelpG and NBO methods. However, there is a debate regarding the choice of the most appropriate partial atomic charges for modeling molecular systems. Therefore, it is essential to discuss the most efficient method that produces the most reliable charge values for molecular modeling [51].

Therefore, it is important to assess the performance of the different methods: Mulliken, ChelpG, and NBO. Selected partial charges were calculated for the parent TUDHIPP molecule, 1H, (Table 2), using six basis sets; namely, B3LYP/6-31G, 6-311G, 6-31+G (d), 6-311+G (d), 6-31++G (d,p) and 6-311++G (d,p). The quality of the charges was estimated from the standard error mean (Figure 4). It follows from these results that the ChelpG gives the smallest standard error mean compared with Mulliken and NBO methods.

### 2.3. Determination of The Chemical Reactivity of TUDHIPP Derivatives in the Gas and Aqueous Phases

#### 2.3.1. Effect of Solvent on Energy and Dipole Moment

The effect of solvents on the properties and behavior of molecules can be determined using various theoretical methods [52]. Our interest is to explore the effect of solvents on the geometrical parameters of TUDHIPPs, which are studied in the gas phase and aqueous phases using the Polarizable Continuum Model (PCM)) solvation model. In this model, the solvent is treated as an unstructured continuum outside the solvent-accessible surface of the solute and it is characterized only by its dielectric constant which is 78.5 for water at 25 °C [53]. Table 3 gives the values of the total energy and dipole moments of compounds 1H-9H, shown in Figure 2 and Figure 5. From Table 3, it is obvious that the total energy decreases in the aqueous phase compared to that in the gas phase making the system more stable in water for all derivatives. For example, for 1H, the energy difference between the molecule in the gas and aqueous phases is (−0.09 eV) which corresponds to the solvation energy of 1H. The dipole moment values for all studied molecules increase in aqueous phase compared to those in the gas phase, indicating that the solvent (water) increases the polarity of the molecules.

#### 2.3.2. Global Reactivity Descriptors

The density functional theory (DFT) defines many important concepts of chemical reactivity using electron density of the chemical system under study [54]. These include *E_HOMO_*, *E_LUMO_*, energy gap (*E_g_*), ionization potential (*I*), electron affinity (*A*), electronegativity (*χ*), chemical hardness (*η*), chemical softness(*S*), chemical potential (*V*), electrophilicity (*ω*) and nucleophilicity (*N*), which describe the global descriptors used to address the various qualitative concepts in chemical reactivity [52,53]. These are calculated at B3LYP/6-311++G (d, p) using the following formula:
*V* = −*χ* =−1/2 (*I* + *A*), *η* = ½ (*I* − *A*), *S* = 1/2*η*, *ω* = (*V*^2^/2*η, N* = *I*_(*TCE*)_ − *I*

Using *HOMO* and *LUMO* energies, the vertical ionization potential energy (*I*) and the vertical electron affinity (*A*) respectively; which can be expressed as *I* = −*E_HOMO_*, *A* = −*E_LUMO_* and *I_TCE_* is the ionization energy of tetracyanoethylene.

It can be shown that these indices measure the propensity of chemical species towards electrons. Thus, a good nucleophile can be characterized by low values of *V* and ω, while a good electrophile can be characterized by high values of *V* and ω [54] where *E_HOMO_* characterizes the electron donating ability, while *E_LUMO_* characterizes the electron withdrawing ability. The energies of *E_HOMO_* and *E_LUMO_* and their neighboring orbitals are all negative, which indicate that all molecules are stable [55], as shown in Table 4.

The energy gap (*E_gap_)* is an important stability index which determines the chemical reactivity of the molecule. From the values reported in Table 4 and Figure 5. it is noticed that the energy gap (*E_gap_*) are small in aqueous phase for derivatives 1H, 2H, 3H, 4H, 5H and 6H (which are known to contain electron withdrawing groups) making these molecules more reactive compared to the same molecules in the gas phase. On the contrary, compounds 7H, 8H and 9H (which are known to contain electron donating groups) have larger *E_g_* in the aqueous phase making them more reactive in the gas phase than in the aqueous phase.

The values reported in Table 4 show that the electronic chemical potential (*V*) and the index of nucleophilicity (*N)* for compounds 1H, 2H, 3H, 4H, 5H and 6H (having electron withdrawing groups) decrease in gas phase compared to those in aqueous phase. On the contrary, compounds 7H, 8H and 9H (having electron donating groups), have high *N* and *V* in the gas phase compared to those in the aqueous phase. Compounds 1H, 2H, 3H, 4H, 5H and 6H (electron withdrawing) have lower nucleophilic nature in aqueous phase than in the gas phase, unlike compounds 7H, 8H and 9H (electron donating groups) have higher nucleophilic nature in the aqueous phase than in the gas phase. The electrophilic nature is high in the gas phase than in the aqueous phase for derivatives 1H, 2H, 3H, 4H, 5H and 6H (electron withdrawing), unlike compounds 7H, 8H and 9H (electron donating) have low electrophilic nature in the gas phase than for the aqueous phase.

Electronegativity (χ), hardness (η) and softness (S) are reported in Table 4, which are considered useful concepts for understanding the behavior of chemical systems. Thus, the global hardness (η) reflects the ability of charge transfer inside the molecule which decreases when the solvent effect is considered, so the molecules 1H, 2H, 3H, 4H, 5H, and 6H are harder in the gas phase than in the aqueous phase. These results may suggest a higher stability of 1H, 2H, 3H, 4H, 5H and 6H compounds in the aqueous phase than in the gas phase. An opposite trend is found for compounds 7H, 8H, and 9H.

#### 2.3.3. Local Reactivity Descriptors

The concepts of local and global reactivity descriptors have been widely used to understand the chemical reactivity and site selectivity [56,57]. To analyze molecular site selectivity, Parr and Yang [58] define local descriptors such as Fukui functions. Thus, calculating Fukui functions can enable us to determine the active sites of a molecule, based on the electronic density changes experienced by the molecule during a reaction. Fukui functions *f^+^* (r), *f^−^* (r) and *f^0^* (r) are calculated for three chemical situations, using the following equations as [59,60,61]:
$$f^{-}\left(r\right)={\;q}_{k}\left(N\right)-q_{k}\left(N\;-1\right)\;\approx {\;\rho }^{\mathrm{HOMO}}\left(r\right),\mathrm{for}\;\mathrm{electrophilic}\;\mathrm{attack}$$
$$f^{+}\left(r\right)={\;q}_{k}\left(N+1\right)-q_{k}\left(N\;\right)\approx \rho^{\mathrm{LUMO}}\left(r\right),\mathrm{for}\;\mathrm{nucleophilic}\;\mathrm{attack}$$
$$f^{0}\left(r\right)=\frac{1}{2}\left[q_{k}\left(N+1\right)-q_{k}\left(N\;-1\right)\right]\approx \frac{1}{2}\left[\rho^{\mathrm{HOMO}}\left(r\right)+\rho^{\mathrm{LUMO}}\left(r\right)\right],\mathrm{for}\;\mathrm{Radical}\;\mathrm{attack}$$
where $q_{k}\left(N\right)\;$is the atomic population on the *k_th_* atom for the neutral molecule, while $q_{k}\left(N+1\right)\;$ and $q_{k}\left(N-1\right)\;$are the atomic population on the *k_th_* atom for its anionic and cationic species, respectively. The values of descriptors calculated at B3LYP/6-311++G (d, p) level using ChelpG charges on atoms for molecule **1H** are presented in Table 5. In addition to the information concerning electrophilic and nucleophilic capacity of a given atomic site in the molecule, Labbe et al. [62] proposed another Dual descriptor $\left(\Delta f\left(r\right)\right)$ which is given by:
$$\Delta f\left(r\right)=f^{+}\left(r\right)-f^{-}\left(r\right)$$where $\Delta f\left(r\right)$ is defined as the difference between the nucleophilic and electrophilic Fukui function. There are two situations need to be considered: if $\Delta f\left(r\right)>0$, then the site is favored for a nucleophilic attack, whereas if $\Delta f\left(r\right)<0$, then the site may be favored for an electrophilic attack. The calculation of Fukui functions indices and Dual descriptor of the parent molecule **1H** obtained from ChelpG charges in the gas and aqueous phase at the level B3LYP/6-311++G (d,p) is given in Table 5. From the values of Fukui functions $f^{-}\left(r\right)\;\mathrm{and}\;f^{+}\left(r\right)$ in the gas phase, it can be stated that the most electrophilic active site is located on C5, S8, N11 and C13. Likewise, the active sites susceptible for nucleophilic attacks are C12, C15, O16 and C17. In the aqueous phase, one can see slight increase of the Fukui functions, which are also located on C5, S8, N11 and C13 for the electrophilic attack. Again, C12, C15, O16 and C17 are favorite sites for the nucleophilic attack. The same conclusion can be reached considering the Dual descriptor $\Delta f\left(r\right)$ regarding electrophilic and nucleophilic attack in both the gas phase and the aqueous phase. These results manifest a strong influence of water as a solvent on the reactivity of the compound considered. 

### 2.4. Molecular Electrostatic Potential Surface (MEPS) of TUDHIPP

There exists a correlation between molecular electrostatic surface potential (MESP) and electronic density, where the latter can be calculated from theoretically derived electron density or from experimentally measured electron density, which has been demonstrated useful in understanding the interaction behavior and chemical properties of molecules [63]. MESP is found to be a useful descriptor to predict reactive sites for electrophilic and nucleophilic attack reactions as well as hydrogen-bonding interactions [64,65].

The MESP map (Figure 6) shows a region characterized by a red color around the oxygen (O10 and O16) atom that refers to a negative potential (electrophilic attack), in additional to that, another region characterized by a blue color around the two hydrogen atoms (H7 and H27) that refers to a positive potential (nucleophilic attack). The green color refers to a neutral electrostatic potential. The information contained in the MESP is used in a variety of different classical and quantum chemical models. Some of the typical applications of the MESP are the interpretation of molecular electronic structure, reactivity, and structure-activity relationships [66].

### 2.5. Molecular Docking Simulations

#### 2.5.1. TUDHIPP Docked with Human DNA Topoisomerase II alpha (II α)

Human topoII is the target of several anticancer agents [67] which includes doxorubicin, amascrine, mitoxantrone and etoposide [68]. Molecular docking of the nine TUDHIPP derivatives with human topoisomerase IIα (4fm9) shows that each TUDHIPP formed hydrogen bonds (HBS) with the active site of topoisomerase IIα as shown in Table 6, Figure 7 and Appendix A.

Compounds 4H, 6H and 7H formed HBs with the ARG929, ASN779 and TYR892 amino acid residues and binding energies were −8.85, −9.17 and −9.23 kcal/mol respectively. 4H and 7H formed another HB with SER778 and GLU854 but 6H formed one with DG10. Compounds 3H and 5H formed HBs with the GLN773, ASN779 and DG10 amino acid residues and the corresponding binding energies were −8.61 and −8.74 kcal/mol respectively. 3H formed another HB with ASN770 and TYR892. Compounds 8H and 9H formed HBs with the GLU854 and TYR892 amino acid residues and the binding energies were -9.06 and -8.53 kcal/mol respectively. 8H formed other HBs with SER778, ILE864 and ARG929. Compound 1H formed HBs with the ASN780 and SER891 amino acid residues and the binding energy was −7.9 kcal/mol, while compound 2H formed HBs with the ASN770, SER800, GLN773 and DC9 amino acid residue and the binding energy was −9.29 (Table 6).

In order to verify the relevance of the derivatives of TUDHIPP as potential antitumor agents we have carried out a molecular docking study of the well-known antitumor etoposide [68]. as shown in Table 6 and Figure 7c. Five HBs are observed in etoposide. The first is assigned to be between NE2 of GLN773 and O1 of etoposide with a length of 2.8 Å. The second is found between O of LYS798 and O9 of etoposide having a length of 3.72 Å. The third is observed between O of SER800 and O9 of etoposide corresponding to a length of 2.81 Å. The fourth found between O3/ DC9.C and O11 of etoposide corresponding to a length of 3.1 Å. The fifth is assigned between O5/ of DG10.C and O11 of etoposide corresponding to a length of 2.64 Å. The binding energy of the five assigned HBs is calculated to be −8.39 kcal/mol. The comparison between the pattern of binding sites and binding energy of etoposide with topoII-α and those of 1H-9H indicate that a good match with 2H, 6H, 5H and 3H. This indicates that TUDHIPPs can be as good candidates as antitumor agents as etoposide.

#### 2.5.2. TUDHIPP docked with human DNA topoisomerase II beta (IIβ)

Molecular docking of nine TUDHIPP derivatives with human topoisomerase IIβ (3Q3X) shows that each TUDHIPP formed a hydrogen bond with the active site of topoisomerase IIβ as shown in Table 7, Figure 8 and Appendix A.

Compounds 1H, 3H and 4H were found to bind with human topoisomerase II b with HBs at the DC8 and DT9 amino acid residue and the binding energies were −7.25, −9.28 and −9.04 kcal/mol respectively. 1H formed HBs with DG10. 2H formed HBs with ASP479, LYS456 and DG10. Compounds 5H, 6H and 9H formed HBs with the ASP479 and DT9 amino acid residue and binding energies were −9.5, −10.07 and −9.16 kcal/mol, respectively. 5H formed another HB with DC8 while 6H formed one with DC11. Compounds 7H and 8H formed HBs with the DC8 amino acid residue and binding energies were −9.34 and −9.56 kcal/mol, respectively. 7H formed another HB with GLN778 while 8H formed one with ASP479 as seen in Table 7.

Similar to the analysis used with topoII-α to verify the validity of the derivatives of TUDHIPP as potential antitumor agents we have carried out molecular docking study of the well-known antitumor etoposide [68]. as shown in Table 7 and Figure 8b. Three interactions with four HBs are observed in etoposide. The first interaction is assigned to be between N, OD1 centers of ASP479 and O9 of etoposide with lengths of 3.08 and 2.78 Å, respectively. The second interaction is found between O3/ of DC8.C and O8 of etoposide having a length of 2.87 Å. The third interaction is observed between O3/ of DT9.D and O9 of etoposide corresponding to a length of 2.81 Å. The fourth found between O3/ DC9.C and O11 of etoposide corresponding to a length of 3.37 Å. The binding energy of the four assigned HBs is calculated to be −11.59 kcal/mol. The comparison between the pattern of binding sites and binding energy of etoposide with topoII-β and those of 1H-9H indicate an even better match with all derivatives except 2H and the highest one is with compound 6H. This indicates that TUDHIPPs can be as good candidates as antitumor agents as etoposide.

### 2.6. Drug Likeness Screening

Drug-likeness is a useful concept as a guide for molecular drug design in a hit and lead optimization [14,69]. In vivo [17,70] pharmacokinetic parameters, such as absorption, distribution, metabolism, and excretion (ADME) are strongly affected by the physicochemical properties of a drug. The analyses of ADME properties are achieved using the rule of thumb [71].

We calculate the parameters of the rules of Lipinski, Veber, Warring and analysis of the Golden Triangle of the nine derivatives of TUDHIPP (Figure 1) with respect to growth inhibition activity of the series of TUDHIPP derivatives which have been synthesized and characterized by Hassaneen et al. [12]. The ADME properties that we will take into account include: number of rotatable bonds (n_rotb_), hydrogen bond donors (HBD), hydrogen bond acceptors (HBA), polar surface area (PSA), Partition coefficient octanol/water (log P), octanol: buffer (pH 7.4) distribution coefficients (logD_7.4_), and molecular weight (MW). Table 8 shows the results which are obtained using Hyper Chem 8.0.7 and Marvin Sketch 18.10.0 softwares. 

These parameters can indicate the ability of drug molecules towards oral absorption or membrane permeability that occurs when these molecules of potential drug likeness follow Lipinski’s rule of five. If a molecule satisfies these rules, then the chance to be considered as a drug is high. When these rules are applied to a certain potential drug molecule; it imposes restrictions on partition coefficient of the molecule between water and octanol, number of hydrogen bond donors (HBD) and hydrogen bond acceptors (HBA), log *P*, and molecular weight (MW). Such rules include these conditions and limits on drug-likeness parameters [34,72]. Compounds which do not satisfy two or more of these rules should not be considered for further development as a drug.

It is obvious that, by referring to Table 8, all compounds satisfy all the rules of five of Lipinski, which demonstrates that the TUDHIPP derivatives have a good absorption and permeation. This can lead us to conclude that these proposed compounds theoretically satisfy the oral bioavailability requirements.

Veber et al. [73] provided two additional descriptors to achieve ideal oral bioavailability. These rules provided by Weber are based on a solid physicochemical basis, where it is well established that the larger the number of hydrogen bonds the lower the solubility in water because these hydrogen bonds should be broken to allow for the permeation of the compounds into and through the cell lipid bilayer membrane [74]. 

There is a relationship between molecular weight (MW) and molecular size, where the larger the molecular size, the larger the cavity formed in water in order to solubilize the compound. An acceptable criterion of the molecular weight of a potential drug molecule is that it should be under 500 DA. Therefore, the smaller the MW of the molecule, the better its absorption is. The results from Table 8 show that the MW of all compounds in the chosen series is below the limit, i.e., 500 DA, indicating an easier passage of these molecules through cell membrane. 

To predict the solubility of oral drug, Log *P* is used as an index, by partitioning the drug molecule between water and *n*-octanol, the hydrophobic solvent, [74]. The results from Table 8 show that Compound (6H) is expected to have the highest hydrophilicity because it has the highest negative log *P* value, this implies that this compound will show a good aqueous-solubility, better gastric tolerance and efficient elimination through the kidneys. On the other hand, compound (5H) will show the most lipophilic character. Consequently, this compound will have good permeability across cell membrane. The results in Table 8 show that all studied compounds having the values of log *P* lower than 5, a requirement by Lipinski rule to satisfy drug-likeness. 

The last index of the Lipinski rules is the number of hydrogen bond donors, HBD (NH and OH) and the number of hydrogen bond acceptors HBA (O and N atoms). This index has shown to be critical in a drug development procedure, since they have an influence on the absorption and permeation [75]. According to the rule set by Lipinski, the number of hydrogen bond donors must be less than 5 and the number of hydrogen bond acceptors must be less than 10. The results from Table 8 show that HBD and HBA of the proposed molecules are found to be within Lipinski’s limit i.e., less than 5 and 10, respectively. It is important to emphasize that molecules violating more than one of these indices set by Lipinski can result in problems with bioavailability and failure to satisfy drug likeness requirements [72,76]. 

In addition to the rules set by Lipinski, Veber proposed another set of rules which considers the number of rotatable bonds (n_rotb_) and Topological Polar Surface Area (TPSA), that need to be considered in order for a molecule to satisfy drug-likeness criteria. The significance of the number of rotatable bonds (n_rotb_) stems from the fact that it measures molecular flexibility, which is considered to be a good parameter of oral bioavailability of drugs [73]. 

In addition to n_rotb_, Topological polar surface area (TPSA) is considered as an important useful parameter for the assessment of drug transport properties. A good correlation was found between TPSA with both the human intestinal absorption, Caco-2 monolayer’s permeability, and blood-brain barrier penetration [77]. Veber set the limits on TPSA that molecules having values of 140 Å^2^ or more are expected to have poor intestinal absorption [78]. The results of TPSA from Table 8 for TUDHIPP derivatives were found in the range of 70.23–113.37 and are well below 140 Å^2^, which shows that these compounds tend to have an intermediate intestinal absorption (Table 8). 

It is obvious that the calculation results from Table 8 show that all compounds meet the rules set by Lipinski and Veber. This suggests that the proposed compounds theoretically would satisfy the requirements for oral bioavailability.

A useful visualization tool of the Golden Triangle developed by Johnson and co-workers, which relates molecular weight (*MW*) and lipophilicity (Log*D* at pH 7.4; log *D*_7.4_). In this sense, the golden triangle is considered as a surrogate of many different molecular descriptors. Compounds that reside inside the Golden Triangle are more likely to be both metabolically stable and to possess good membrane permeability than those outside. The Golden Triangle (Figure 9) shows that the compounds 3H, 5H, 6H and 9H are located outside the triangle; therefore these derivatives have poor permeability and clearance. On the other hand, compounds 1H, 2H, 4H, 7H, and 8H are located inside the triangle, accordingly, they have better permeability and clearance [69]. In general; molecules with lower log D and higher molecular weight fail to satisfy drug-likeness as a result of their low permeability. On the other hand, molecules with higher log *D* and higher MW fail to satisfy drug-likeness, due to elevated in vitro clearance [79].

We have compared the results of Lipinski rule of five and Veber’s rule of TUDHIPP versus the very well-known antitumor drug etoposide to assess where TUDHIPPs stand as anti-tumor drugs. We have found that the TUDHIPPs satisfy all Lipinski and Veber’s rules, while etoposide satisfies only half of Liipinski and Veber’s rules, indicating the superiority of the TUDHIPPs over etoposide as an antitumor drug from theoretical point of view. One of the obstacles of using etoposide as an antitumor drug is its sparing solubility in cell medium, therefore the TUDHIPP solubility results suggest that they can be better antitumor drugs than etoposide. 

We have computed the descriptors of Lipinski and Veber’s rules for etoposide. The comparison of our computed results with the published data on etoposide, as given in Table 8, shows a good agreement between our computed descriptors and those published. The comparison of our computed descriptors of Lipinski rules of etoposide and those of TUDHIPPs shows that the logP of 2H is the closest to that of etoposide, the MW of 9H is the closest to etoposide, the HBDs of all TUDHIPPs except 8H are the same as those of etoposide, the HBAs of TUDHIPPs satisfy the rule while etoposide violates that rule. A similar comparison of Veber’s rule between the TUDHIPPs and etoposide shows that for n_rot_, both satisfy the rule. For PSA, etoposide does not obey the rule, while TUDHIPPs obey the rule and the closest TUDHIPP derivative is 6H. For logD_7.4_, the closest value to that of etoposide is that of 1H.

### 2.7. Structural Activity/Property Relationship

This section discusses the calculation of the physicochemical properties of TUDHIPP derivatives, as shown in Table 9. These physicochemical properties considered are molar volume (*V*), hydration energy (*HE*), molar refractivity (*MR*), surface area grid (*SAG*) and polarizability (*Pol*). The calculations were performed using HyperChem 8.0.7

Molecular polarizability depends only on molecular volume, where the latter determines transport characteristics of molecules, which includes blood-brain barrier penetration and intestinal absorption. Therefore, modeling of molecular properties and biological activity requires the use of molecular volume in QSAR studies [80].

There is another SAR parameter that is the molar refractivity (*MR*), from the results obtained in Table 9, we have noticed that, polarizability data, molecular refractivity and surface area grid are generally proportional to the size (*V*) and the molecular weight of TUDHIPP derivatives. For instance compound number 9H which has the highest value of volume (1070.91 (Å^3^), shows the maximum values of polarizability (45.9 (Å^3^), refractivity (116.3 (Å^3^) and surface area grid (621.57 (Å^3^). On the other hand, compounds 5H, 6H which have lower values of molecular volume (1000.77 Å^3^) and 998.3 Å^3^, respectively, than that of 9H, still have high polarizbility (43.51 Å^3^), 42.72 Å^3^, respectively, refractivity (109.49 Å^3^ and 108.18 Å^3^, respectively, and surface area grid (590.22 Å^3^ and 586.26 Å^3^, respectively, but not higher than that of compound 9H. Although compound 9H, which has the highest molecular volume (1070.91 Å^3^) does not have the highest molecular weight MW (402.74), however, compounds 5H and 6H which have lower values of molecular volume, (1000.77 Å^3^, and 998.3 Å^3^, respectively, than that of 9H (1070.91 Å^3)^, they have the largest value of molecular weight (438.3 amu), i.e., there does not exist a correspondence between molecular volume and molecular weight.

The results in Table 9 show that an increase in the hydrophobic values resulting in a decrease of hydration energy. The stability of the different molecular conformations in water solutions is determined from the hydration energy [81]. The change in the values of the hydration energy is affected by the increase or the decrease in the number of hydrogen bond donor and acceptor. Table 9 illustrates that compound 6H has the highest value of hydration energy in absolute value (−16.69 *kcal/mol*) and characterized by high value of hydrogen bonds donor (3) and acceptor (6). 

Lipophilicity is a major determinant of many ADME properties. LogP expresses the portioning of the drug molecules between aqueous medium outside the cell membrane and the lipid nature of the cell membrane. This means that compounds with a lower logP, are more polar and have poorer lipid bilayer permeability, while compounds with a higher logP are more nonpolar and have poor aqueous solubility [82]. For that, compound 6H has a good aqueous solubility and a bad absorption and permeability and vice versa for compound 5H. Furthermore, the logP values of compounds 2H, 3H, 4H, 5H, 7H, 8H and 9H are in the range of optimal values (0 < logP < 3) [34]. From all the above, one can state that these compounds have a good oral bioavailability and an optimal biological activity.

When we consider the actual biological environments, where the polar drug molecules are surrounded by water molecules, there is a possibility of hydrogen bond formation between water molecules and these drug molecules. The mechanism of hydrogen bond formation requires the interaction between the proton donor sites of the drug molecule with the oxygen atom of water. A mutual interaction is also possible between the acceptor sites of the drug molecule with the hydrogen atom of water. It is important to note, from Table 8, that the hydration energy decreases when the drug molecules contain hydrophobic moieties as in 1H-5H, 7H, and 9H of TUDHIPP. While in the opposite case, the presence of hydrophilic groups as in compounds (6H and 8H), shown in Figure 10 for 6H, having three (HBD):(3 NH) and six (HBA):(four O, 1 N and 1 S) leads to the increase of the hydration energy.

We have computed the QSAR descriptors for etoposide as shown in Table 9, which were compared with other published data on etoposide. The results of the published data on etoposide in Table 9 show a good agreement with our computed Pol and refractivity descriptors. For example, for compound 9H, the computed polarization and refractivity give values of 45.9 Å^3^ and 116.3 Å^3^, respectively, indicating a good agreement with computed polarization and refractivity of etoposide, giving 55.15 Å^3^, and 138.73 Å^3^, respectively. The last two rows of Table 9 indicate clearly the agreement between our computed descriptors and published descriptors of etoposide. These results indicate that the computed QSAR descriptors of the TUDHIPPs is comparable to those of etoposide which suggests that TUDHIPPs can be considered as effective antitumor drugs as etoposide. 

## 3. Computational Details

### 3.1. Geometry Optimization

In this study, all computations were carried out using the Gaussian 09W software package [83]. The molecular geometry for the compound was fully optimized using three different methods. Hartree-Forck (HF) [84], density functional theory with the Becke’s three parameter exchange functional and the gradient corrected functional of Lee, Yang and Parr (DFT/B3LYP) [85,86,87,88] and Möller-Plesset second order (MP2) level [84]. With two basis set 6-311++G (d, p) [89], and cc-pVDZ [90]. No symmetry constrains were applied during the geometry optimization [91,92]. The vibrational frequencies of each molecule have been determined at the same level of theory and it has been checked that all structures correspond to true minima of the potential energy surface.

In this study, we use DFT to define molecular stability and reactivity descriptors, to determine a reactive site of the molecule. We have calculated the Fukui function [56,57,58,59,60,61] and Dual descriptor [62] as a descriptor of reactivity followed by the visualization of MESP using GaussView 5.0.9 [93] and optimized geometry visualized using Chemcraft version 1.6 package [94].

Finally, these previous geometry optimized structures are saved to use in next computation. The SAR properties were obtained from the module QSAR properties involved in HyperChem software version 8.0.7 [95]. Using the Calculator Plugins of MarvinSketch 18.10.0 [96] software, the calculated parameters of drug-likeness according to Lipinski rule (the rule of five) [72], and Veber’s rule [73] are obtained. 

### 3.2. Molecular Docking Simulation

This section includes four parts; ligand structure preparation, protein structure preparation, ligand –protein docking, and molecular docking analysis:

#### 3.2.1. Ligand Structure Preparation

The most suitable previously geometry optimized structures of 1H-9H of TUDHIPP at *B3LYP/6-311++G* (*d, p*) method of calculations are saved in pdb file format. Such Pdb files converted to PDBQT files format using Autodock Tools version 1.5.6rc3 [97,98].

#### 3.2.2. Protein Structure Preparation

In order to prepare the protein structures for molecular docking simulation, several sequential steps need to be followed. The first step requires the definition of the appropriate protein for docking with the proposed molecules, **1H-9H**. In this case, human DNA topoisomerase II α (PDB ID: 4FM9) [99] and β (PDB ID: 3QX3) [100], were chosen and were retrieved from PDB (http://www.rcsb.org/pdb/). This is followed by cleaning both proteins from heteroatoms, followed by molecular mechanics energy minimization using Swiss-PdbViewer [101]. At that point both clean proteins are ready for the next step, which involves the generation of the docking input file using Autodock Tools version 1.5.6rc3. The third step requires the addition of polar hydrogen atoms, which will be achieved using AutoDock Tools (ADT) 1.5.6rc3, where the Kollman charges were added, AD4 type atoms were assigned and the protein saved in PDBQT file format. The fourth step involves the preparation of the grid and docking parameter files using ADT, where the molecular docking studies were performed using AutoDock 1.5.6rc3 [97,98], by considered all the bonds of ligands of 1H–9H as rotatable and those of the topo II α, and β as rigid [97]. We selected a grid box size of 90 × 90 × 90 Å with 0.375 Å spacing centered at the site of DNA cleavage of topo–DNA complexes.

#### 3.2.3. Molecular Docking

In order to perform Molecular docking study of TUDHIPP derivatives versus Human DNA Topoisomerase II, we used AutoDock1.5.6rc3^®^ suite [97,98]. The molecular docking parameters include the Lamarckian Genetic Algorithm (LGAs), mutation rate, and cross over rate. For the Lamarckian Genetic Algorithm, we have chosen initial population size equal to 150 and maximum number of evals as 2500 000 energy evaluations. We used a rate of gene mutation of 0.02 and a crossover rate of 0.80. One hundred possible binding conformations, was generated using AutoDock1.5.6rc3^®^ tools. Pre calculated grid maps obtained by Autogrid [98] were used to obtain the docking simulation.

#### 3.2.4. Molecular Docking Analysis

After the completion of the docking of the ligand-protein complex using AutoDock1.5.6rc3^®^, we select the conformers with lowest binding free energy. These docked low free energy protein–ligand structures, were saved as PDBQT files. We analyzed the van der Waal and hydrogen bonds interactions between TUDHIPP derivatives and topo II enzyme using LIGPLOT^+^ program [102]. We captured and saved the nine docked structures indicating such interactions.

## 4. Conclusions

This work provides detailed insights into the electronic structure properties and its link to drug-like properties and structure-activity relationships of TUDHIPPs using various computational approaches such as the HF, MP2, and DFT/B3LYP methods by using cc-pVDZ and 6-311++G (d,p) as basis sets. Among the various computational methods, the B3LYP/6-311++G (d,p) is found to reproduce experimental geometrical results and is used for further computational investigations on TUDHIPPs. Calculation of atomic charges using different methods is carried out followed by the calculation of reactivity descriptor of TUDHIPPs in the gas and aqueous phases. We found that the solvation modifies the values of the reactivity descriptors as a result of the interaction between solvent and the TUDHIPP molecules. A noticeable effect of water on the reactivity of TUDHIPPs is observed, as reflected on the values of the Dual descriptor and Fukui function. The computational results predict that compound 9H is the most reactive compound having the smallest energy gap of all studied compounds (1H-9H). The results also show that C5, N11 and C13 are the most preferred sites for electrophilic attack. The results of molecular electrostatic potential surface give useful information on the binding sites of TUDHIPPs as drugs to the protein actives sites.

The molecular docking studies revealed that TUDHIPP exhibits molecular interactions with human topoisomerase II α and II β. In order to evaluate TUDHIPPs as anticancer drugs, we have carried out a molecular docking study of the well-known anticancer etoposide with human topoisomerase II α and II β. Compound 2H shows the lowest binding energy of -9.29 kcal/mol with topoisomerase II α. Compound 6H showed binding energy of -10.07 kcal/mol with topoisomerase II β. The TUDHIPP derivatives show good binding scores in molecular docking studies with Topo II α and II β enzymes compared to those of the well-known anticancer Etoposide, indicating a potential anticancer activity of TUDHIPPs, which need further in vitro and in vivo studies of these derivatives.

The drug-likeness and QSAR descriptors show that, compound 5H is expected to have the highest hydrophilicity, whereas compound 6H will be the most lipophilic. This implies that these compounds will have poor permeability across cell membranes. Compound 6H has important hydration energy. The application of Lipiniski, Vener’s rules and Golden Triangle analyses on the studied TUDHIPP derivatives shows that these compounds theoretically will not have problems with the oral bioavailability, and most of them are more likely to be both metabolically stable and to possess a good membrane permeability. For a drug to be considered as a successful drug, a delicate balance of many factors relating to their biological and physicochemical properties is required. It is clear that most of the proposed compounds in this study satisfy this requirement. The comparison of the drug likeness descriptors of TUDHIPPs with those of etoposide, which is known to be an antitumor drug, indicates that TUDHIPPs can be considered antitumor drugs. The overall study indicates that TUDHIPPs have comparable and even better descriptors than etoposide suggesting they can be potential antitumor drugs.

## Figures and Tables

**Figure 1 ijms-21-01253-f001:**
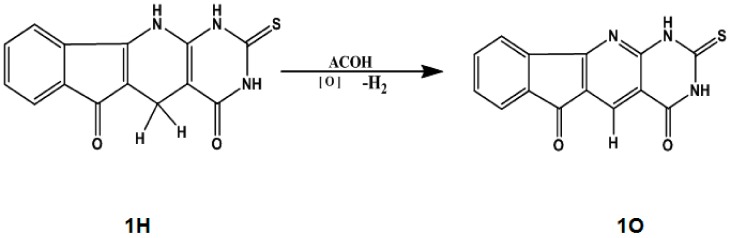
The oxidation of 1H form of TUDHIPP to 1O.

**Figure 2 ijms-21-01253-f002:**
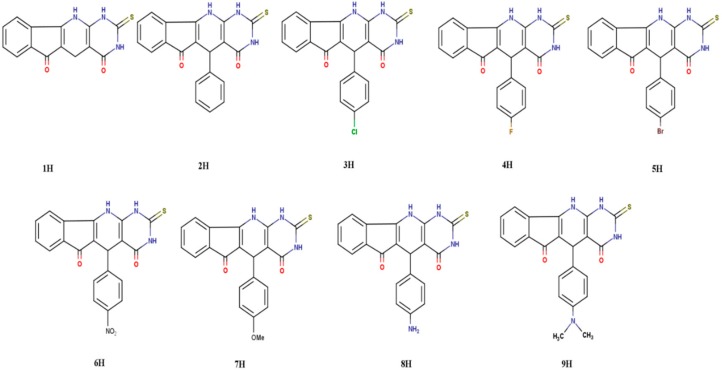
2D structure of the thiouracil-based dihydroindenopyridopyrimidine (TUDHIPP) derivatives 1H–9H.

**Figure 3 ijms-21-01253-f003:**
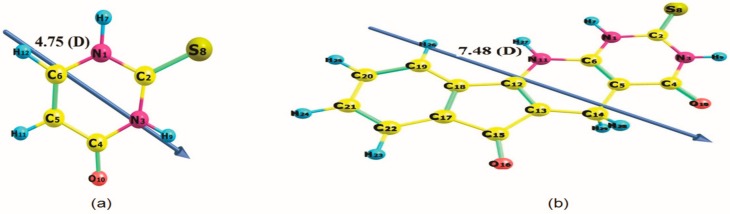
Optimized geometry, numbering system, vector of dipole moment, value of dipole moment (D) of (**a**) thiouracil nucleus and (**b**) compound 1H at B3LYP/6-311++G (*d*, *p*) level of calculation.

**Figure 4 ijms-21-01253-f004:**
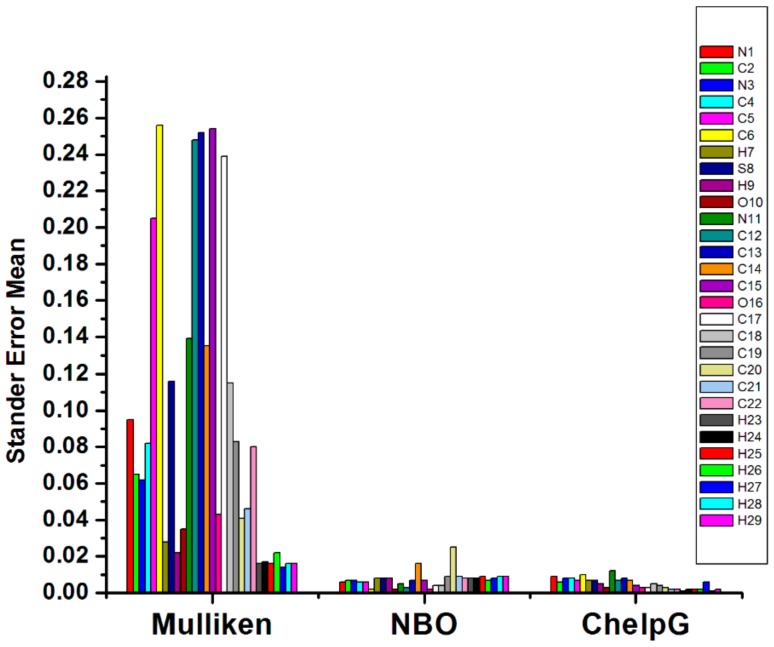
Standard error mean of partial charges of compound 1H using B3LYP level of theory with different basis sets.

**Figure 5 ijms-21-01253-f005:**
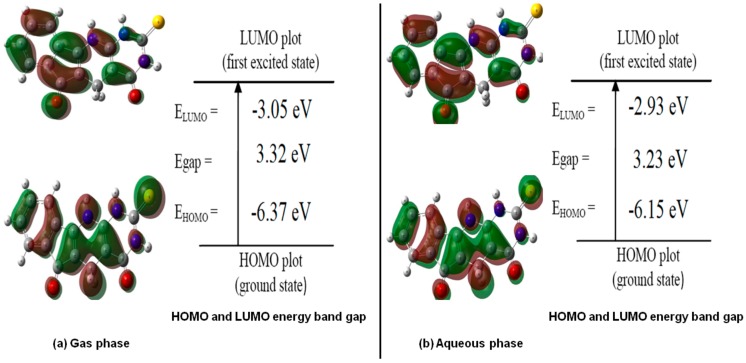
Highest occupied molecular orbital (HOMO) and lowest occupied molecular orbital (LUMO) of compound 1H: (**a**) in gas and (**b**) aqueous phases.

**Figure 6 ijms-21-01253-f006:**
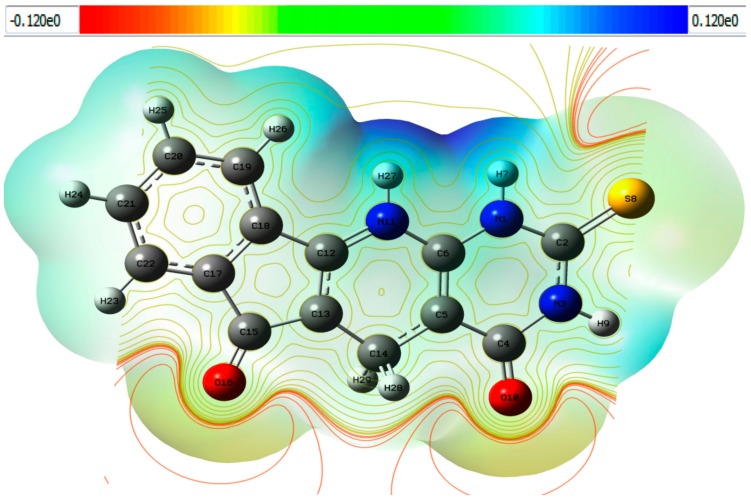
2D and 3D Molecular electrostatic potential map of compound 1H of TUDHIPP derivatives.

**Figure 7 ijms-21-01253-f007:**
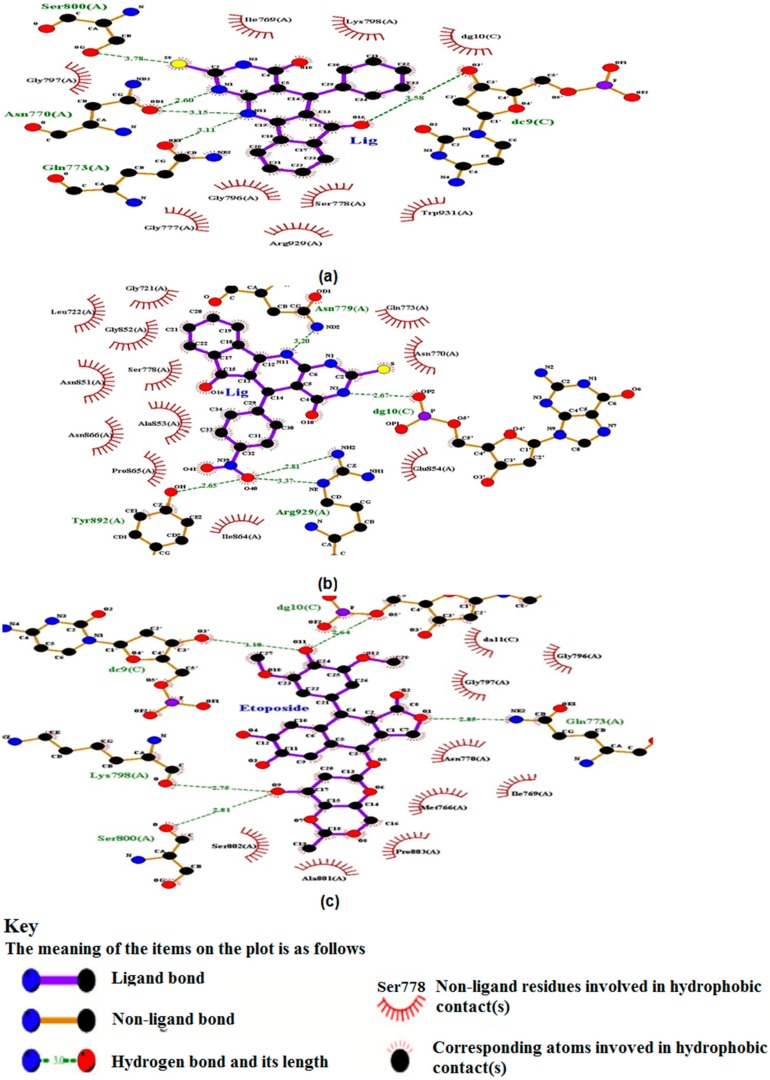
(**a**–**c**) Two-dimensional binding sites scheme of 2H, 6H and etoposide, respectively, with human DNA topoisomerase IIα, 4fm9.

**Figure 8 ijms-21-01253-f008:**
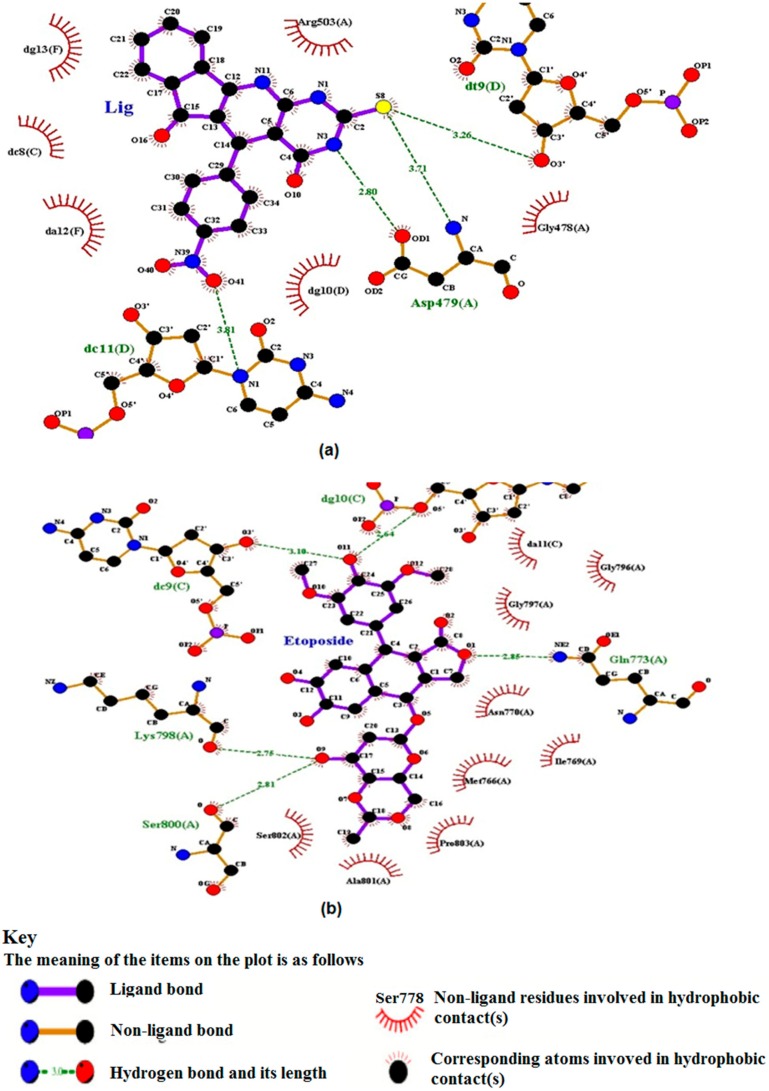
(**a**,**b**) Two-dimensional binding sites scheme of 6H and Etoposide respectively with human DNA topoisomerase IIβ, 3qx3.

**Figure 9 ijms-21-01253-f009:**
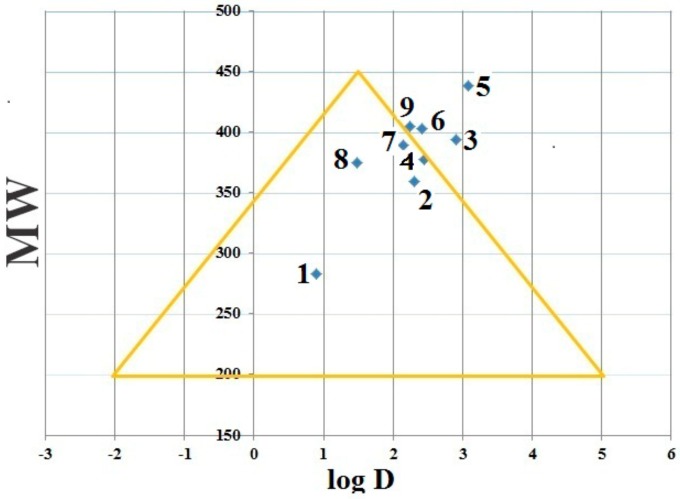
Golden triangle of DHIPP, **1–9** compounds **1H–9H** respectively.

**Figure 10 ijms-21-01253-f010:**
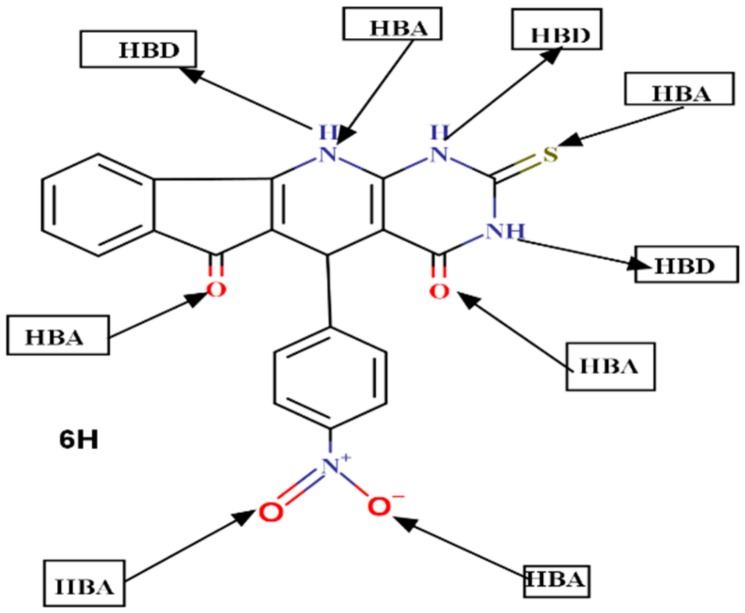
Donor and acceptor sites of compound 6H.

**Table 1 ijms-21-01253-t001:** Experimental bond length and computed values of thiouracil (TU) nucleus at different levels of calculations.

Parameters	Exp.*	6-311++G (d, p)			CC-PVDZ		
		ab initio/HF	ab initio/MP2	ab initio/B3LYP	ab initio/HF	ab initio/MP2	ab initio/B3LYP
R(N1,C2)	1.351	1.353	1.380	1.378	1.355	1.382	1.380
R(N1,C6)	1.371	1.374	1.376	1.374	1.374	1.377	1.375
R(N1,H7)	1.030	0.994	1.011	1.010	0.998	1.016	1.014
R(C2,N3)	1.357	1.350	1.376	1.370	1.351	1.376	1.370
R(C2,S8)	1.683	1.661	1.643	1.662	1.665	1.658	1.668
R(N3,C4)	1.394	1.398	1.414	1.417	1.398	1.418	1.419
R(N3,H9)	1.030	0.998	1.015	1.013	1.002	1.021	1.018
R(C4,C5)	1.442	1.460	1.458	1.456	1.462	1.463	1.460
R(C4,O10)	1.235	1.187	1.220	1.214	1.190	1.222	1.217
R(C5,C6)	1.354	1.328	1.356	1.348	1.332	1.362	1.353
R(C5,H11)	1.080	1.070	1.082	1.079	1.077	1.090	1.088
R(C6,H12)	1.080	1.073	1.085	1.083	1.080	1.093	1.091
A%		1.386	1.200	1.102	1.192	1.257	1.105

$A\%=\frac{\mathrm{Mean}\;\mathrm{absolute}\;\mathrm{deviation}\;\mathrm{of}\;\mathrm{bond}\;\mathrm{length}}{\mathrm{Mean}\;\mathrm{bond}\;\mathrm{length}\;\mathrm{of}\;\mathrm{experemential}\;\mathrm{values}\;}\;\times \;100;$ * Ref. [49,50,51].

**Table 2 ijms-21-01253-t002:** Selected calculated partial charges of compound 1H using DFT (B3LYP functional) at different basis set.

B3lyp	Atoms	6−31G	6−311G	6−31+G (d)	6−311+G (d)	6−31++G (d,p)	6−311++G (d,p)	Stander Error Mean
**Mulliken**	N1	−0.742	−0.774	−0.572	−0.377	−0.393	−0.175	0.095
	C2	0.215	0.160	0.251	0.528	0.174	0.476	0.065
	N3	−0.674	−0.703	−0.612	−0.525	−0.393	−0.330	0.062
	C4	0.462	0.554	0.919	0.699	0.766	0.383	0.082
	C5	0.064	−0.165	0.293	0.872	0.308	1.170	0.205
	C6	0.628	0.811	0.078	−0.334	−0.378	−0.809	0.256
	S8	−0.064	−0.058	−0.068	−0.580	0.011	−0.600	0.116
	O10	−0.427	−0.378	−0.513	−0.327	−0.487	−0.296	0.035
	N11	−0.844	−0.916	−0.599	−0.328	−0.320	−0.036	0.139
	C12	0.250	0.286	−0.318	−0.871	−0.560	−1.235	0.248
	C13	0.072	−0.007	1.429	1.339	0.737	1.004	0.252
	C14	−0.309	−0.414	−0.594	−1.241	−0.549	−0.765	0.135
	C15	0.190	0.234	−0.651	−1.284	−0.320	−1.004	0.254
	O16	−0.387	−0.317	−0.479	−0.247	−0.468	−0.239	0.043
**NBO**	N1	−0.587	−0.573	−0.608	−0.583	−0.607	−0.587	0.006
	C2	0.269	0.277	0.239	0.269	0.239	0.269	0.007
	N3	−0.613	−0.588	−0.634	−0.608	−0.633	−0.613	0.007
	C4	0.654	0.619	0.654	0.654	0.654	0.654	0.006
	C5	−0.195	−0.165	−0.205	−0.193	−0.202	−0.195	0.006
	C6	0.429	0.417	0.420	0.430	0.420	0.429	0.002
	S8	−0.177	−0.224	−0.169	−0.180	−0.170	−0.177	0.008
	O10	−0.592	−0.584	−0.596	−0.593	−0.599	−0.592	0.002
	N11	−0.576	−0.564	−0.596	−0.574	−0.595	−0.576	0.005
	C12	0.225	0.220	0.236	0.236	0.233	0.225	0.003
	C13	−0.144	−0.113	−0.162	−0.144	−0.154	−0.144	0.007
	C14	−0.392	−0.410	−0.478	−0.397	−0.462	−0.392	0.016
	C15	0.540	0.498	0.550	0.531	0.537	0.540	0.007
	O16	−0.523	−0.512	−0.526	−0.524	−0.526	−0.523	0.002
**ChelpG**	N1	−0.395	−0.423	−0.370	−0.386	−0.370	−0.362	0.009
	C2	0.305	0.321	0.286	0.308	0.284	0.297	0.006
	N3	−0.377	−0.400	−0.354	−0.384	−0.350	−0.362	0.008
	C4	0.661	0.709	0.670	0.700	0.672	0.688	0.008
	C5	−0.447	−0.487	−0.447	−0.476	−0.461	−0.478	0.007
	C6	0.520	0.561	0.511	0.509	0.548	0.503	0.010
	S8	−0.413	−0.407	−0.380	−0.381	−0.379	−0.379	0.007
	O10	−0.555	−0.578	−0.568	−0.571	−0.570	−0.569	0.003
	N11	−0.598	−0.619	−0.601	−0.572	−0.648	−0.575	0.012
	C12	0.214	0.213	0.207	0.202	0.247	0.228	0.007
	C13	−0.398	−0.422	−0.368	−0.368	−0.383	−0.388	0.008
	C14	0.539	0.578	0.558	0.555	0.538	0.574	0.007
	C15	0.559	0.587	0.565	0.560	0.566	0.562	0.004
	O16	−0.489	−0.506	−0.508	−0.504	−0.508	−0.503	0.003

**Table 3 ijms-21-01253-t003:** Total electronic energy *au,* and dipole moment *D,* of the TUDHIPP derivatives 1H–9H at B3lyp/6-311++G (d, p) level of calculation in gas and aqueous.

Compound	Gas Phase		Water Phase	
	*E_T_*, au	*μ,* D	*E_T_*, au	*μ, D*
1H	−1252.08722	7.48	−1252.11530	11.01
2H	−1483.19061	6.96	−1483.22089	11.02
3H	−1942.81366	7.99	−1942.84419	12.23
4H	−1582.45943	7.86	−1582.49025	12.04
5H	−4056.73331	8.01	−4056.76391	12.24
6H	−1687.75484	10.68	−1687.79006	15.26
7H	−1597.74680	7.78	−1597.77917	12.19
8H	−1538.56733	6.98	−1538.60157	11.09
9H	−1617.19402	6.59	−1617.22616	10.44

**Table 4 ijms-21-01253-t004:** Global reactivity descriptors of the TUDHIPP derivatives (1H–9H), at B3lyp/6-311++G (*d*, *p*) level of calculation in gas and aqueous phases.

Comp.	Phases	*E_HOMO_*, *au*	*E_LUMO_, au*	*E_g_*, *eV*	*I, eV*	*A, eV*	*χ, eV*	*η, eV*	*S, eV* ^−1^	*V, eV*	*ω, eV*	*N, eV*
**1H**	Gas phase	−0.23393	−0.11200	3.32	6.37	3.05	4.71	1.66	0.30	−4.71	6.68	−2.15
	Aqueous phase	−0.22618	−0.10754	3.23	6.15	2.93	4.54	1.61	0.31	−4.54	6.39	−1.94
**2H**	Gas phase	−0.23738	−0.11034	3.46	6.46	3.00	4.73	1.73	0.29	−4.73	6.47	−2.25
	Aqueous phase	−0.23132	−0.10799	3.36	6.29	2.94	4.62	1.68	0.30	−4.62	6.35	−2.08
**3H**	Gas phase	−0.24158	−0.11415	3.47	6.57	3.11	4.84	1.73	0.29	−4.84	6.76	−2.36
	Aqueous phase	−0.23243	−0.10867	3.37	6.32	2.96	4.64	1.68	0.30	−4.64	6.40	−2.11
**4H**	Gas phase	−0.24081	−0.11339	3.47	6.55	3.09	4.82	1.73	0.29	−4.82	6.70	−2.34
	Aqueous phase	−0.23216	−0.10839	3.37	6.32	2.95	4.63	1.68	0.30	−4.63	6.37	−2.11
**5H**	Gas phase	−0.24176	−0.11435	3.47	6.58	3.11	4.85	1.73	0.29	−4.85	6.77	−2.37
	Aqueous phase	−0.23248	−0.10874	3.37	6.33	2.96	4.64	1.68	0.30	−4.64	6.40	−2.12
**6H**	Gas phase	−0.24851	−0.12131	3.46	6.76	3.30	5.03	1.73	0.29	−5.03	7.31	−2.55
	Aqueous phase	−0.23470	−0.11624	3.22	6.39	3.16	4.77	1.61	0.31	−4.77	7.07	−2.18
**7H**	Gas phase	−0.22399	−0.10882	3.13	6.09	2.96	4.53	1.57	0.32	−4.53	6.54	−1.88
	Aqueous phase	−0.22796	−0.10734	3.28	6.20	2.92	4.56	1.64	0.30	−4.56	6.34	−1.99
**8H**	Gas phase	−0.20975	−0.10691	2.80	5.71	2.91	4.31	1.40	0.36	−4.31	6.63	−1.50
	Aqueous phase	−0.21423	−0.10680	2.92	5.83	2.91	4.37	1.46	0.34	−4.37	6.53	−1.62
**9H**	Gas phase	−0.19625	−0.10520	2.48	5.34	2.86	4.10	1.24	0.40	−4.10	6.79	−1.13
	Aqueous phase	−0.20005	−0.10646	2.55	5.44	2.90	4.17	1.27	0.39	−4.17	6.83	−1.23

**Table 5 ijms-21-01253-t005:** Values of the Fukui functions and Dual descriptor of compound **1H** of TUDHIPP at B3lyp/6-311++G (d,p) level of calculation in gas and aqueous phases, where bold number is the significance values in table.

Atoms	Gas Phase			Water		
	*f^(−)^*	f^(+)^	*Δf*	*f* ^(−)^	*f^(+)^*	*Δf*
N1	0.019	0.006	−0.013	0.005	0.007	0.002
C2	0.006	0.004	−0.002	0.001	0.006	0.005
N3	0.018	0.009	−0.010	0.010	0.012	0.002
C4	0.009	0.011	0.001	0.012	0.015	0.003
C5	**0.165**	0.041	**−0.124**	**0.129**	0.044	**−0.085**
C6	0.041	0.035	−0.005	0.025	0.043	0.017
H7	0.000	0.000	0.000	0.000	0.000	0.000
S8	**0.120**	0.005	**−0.115**	**0.051**	0.005	**−0.046**
H9	0.000	0.000	0.000	0.000	0.000	0.000
O10	0.021	0.011	−0.010	0.022	0.012	−0.010
N11	**0.107**	0.015	**−0.092**	**0.118**	0.017	**−0.101**
C12	0.060	**0.155**	**0.095**	0.063	**0.168**	**0.105**
C13	**0.178**	0.077	**−0.101**	**0.241**	0.058	**−0.183**
C14	0.031	0.001	−0.030	0.029	0.001	−0.028
C15	0.008	**0.148**	**0.141**	0.014	**0.166**	**0.152**
O16	0.021	**0.154**	**0.133**	0.030	**0.143**	**0.113**
C17	0.026	**0.090**	**0.064**	0.038	**0.078**	**0.040**
C18	0.024	0.069	0.045	0.036	0.061	0.025
C19	0.011	0.024	0.012	0.013	0.025	0.012
C20	0.023	0.058	0.035	0.034	0.055	0.021
C21	0.032	0.061	0.029	0.045	0.053	0.008
C22	0.004	0.025	0.021	0.005	0.030	0.025
H23	0.000	0.000	0.000	0.000	0.000	0.000
H24	0.000	0.001	0.000	0.000	0.001	0.000
H25	0.000	0.001	0.000	0.000	0.001	0.000
H26	0.000	0.000	0.000	0.000	0.000	0.000
H27	0.002	0.000	−0.002	0.002	0.000	−0.002
H28	0.039	0.000	−0.039	0.038	0.000	−0.038
H29	0.036	0.000	−0.035	0.038	0.000	−0.038

**Table 6 ijms-21-01253-t006:** Molecular interactions of the nine derivatives of TUDHIPP (1H−9H) with human-DNA topoisomerase II α (4fm9).

Ligand	Run no.	Interaction Residue in Human Topo II	Interaction Atoms (Amino Acid…Ligand) HB Length (A^o^)	H Bonds Formed	Binding Energy *Kcal/mol*	Inhibition Constant K_i_, nM
1H	22	ASN780.A SER891.A	ND2……….O16 (3.06) OG…………H9N3 (2.71)	2	−7.9	1.61, µM
2H	18	GLN773.A ASN770.A SER800.A DC9.C	OE1………..H27N11 (3.11) OD1………..H7N1 (2.6), OD1…………H27N11 (3.15) OG…………S8 (3.78) O3…………O16 (3.58)	5	−9.29	155.7
3H	46	GLN773.A ASN770.A ASN779.A TYR892.A DG10.C	OE1………...H7N1 (3.7) O…………...H7N1 (3.72) ND2………..H27N11 (3.17) OH…………Cl39 (3.16) OP2………..N3H9 (2.65)	5	−8.61	485.91
4H	62	SER778.A GLU854.A ARG929.A ASN779.A TYR892.A	OG…………H7N1 (3.19) N…………...O16 (2.98) NH2………..O10 (2.85) OD1………..H27N11 (2.62) OH…………F39 (2.66)	5	−8.85	327.53
5H	98	GLN773.A ASN779.A DG10.C	OE1………..H7N1 (3.69) ND2……….H27N11 (3.2) OP2………..H9N3 (2.62)	3	−8.74	392.18
6H	3	DG10.C ARG929.A TYR892.A ASN779.A	OP2………..N3H9 (2.67) NE…………O40 (3.37), NH2…………O40 (2.81) OH…………O40 (2.65) ND2...............H27N11 (3.2)	5	−9.17	189.18
7H	48	SER778.A GLU854.A TYR892.A ARG929.A ASN779.A	OG…………H7N1 (3.18) N…………...O16 (2.99) OH…………O39 (2.69) NH2………..O10 (3.17) OD1………..H27N11 (2.6)	5	−9.23	170.11
8H	43	SER778.A GLU854.A TYR892.A ILE864.A ARG929.A	OG…………H7N1 (3.02) N…………...O16 (3.02) OH…………N39 (2.65) O…………...H40N39 (2.81) NH2………..O10 (3.23)	5	−9.06	227.858
9H	93	TYR892.A GLU854.A	OH…………S8 (3.121) N…………...O10 (3.05)	2	−8.53	555.3
Etoposide	43	GLN773.A LYS798.A SER800.A DC9.C DG10.C	NE2………..O1 (2.85) O………….. O9 (2.75) O…………...O9 (2.81) O3^/^…………O11 (3.1) O5^/^…………O11 (2.64)	5	−8.39	708.84

**Table 7 ijms-21-01253-t007:** Molecular interactions of the nine derivatives of TUDHIPP (1H–9H) with human DNA topoisomerase II*β* (3qx3).

Ligand	Run no.	Interaction residue in human Topo IIα	Interaction Atoms (Amino Acid …Ligand) HB Length (A^o^)	H Bonds Formed	Binding Energy *Kcal/mol*	Inhibition Constant K_i_, nM
1H	56	DC8.C DG10.D DT9.D	O3…………O16 (3.74) O4/…………H7N1 (3.05) O3/…………H27N11 (2.89)	3	−7.25	4.88, µM
2H	4	ASP479 LYS456 DG10.D	OD1………..H7N1 (3.05), OD1…H27N11 (3.13) NZ…………S8 (3.17) O3/…………S8 (3.19)	4	−9.29	154.3
3H	31	DC8.C DT9.D	O3/…………S8 (3.13) OP2………...H9N3 (2.71)	2	−9.28	157.67
4H	71	DC8.C DT9.D	O3/…………S8 (3.31) OP2………...H9N3 (2.66)	2	−9.04	235.79
5H	75	ASP479.A DC8.C DT9.D	OD1……….H7N1 (2.81), OD1….H27N11 (3.15) O3/………...O16 (3.11) O3/………...H27N11 (3.3)	4	−9.5	108.42
6H	28	ASP479 DC11.D DT9.D	OD1……….H9N3 (2.8), N…S8 (3.71) N1…………O41 (3.81) O3/…………S8 (3.26)	4	−10.07	41.62
7H	75	GLN778.A DC8.C	N…………...O10 (3.51) O3/…………S8 (3.58)	2	−9.34	142.45
8H	28	ASP479.A DC8.C	OD1………..N39 (2.64), N…N39 (3.12) O3/…………O10 (3.39), N1…O10 (3.46)	4	−9.56	97.78
9H	49	ASP479.A DT9.D	OD1……….H7N1 (3.27), OD1…H27N11 (3.04) O3/…………H27N11 (3.42)	3	−9.16	193.43
Etoposide	19	ASP479.A DC8.C DT9.D	N…………...O9 (3.08), OD1….O9 (2.78) O3/…………O8 (2.87) O3/…………O9 (3.37)	4	−11.59	3.17

**Table 8 ijms-21-01253-t008:** Pharmacological activities and properties using Multi Parameter Optimization (MPO) method for the nine TUDHIPP derivatives (1H–9H).

Compounds	Lipinski Rules					Veber Rules		Log D (7.4)
	Log P	MW (DA)	HBD	HBA	Lipinski Score of 5	n_rotb_	PSA	
**1H**	−0.16	283.3	3	4	4	0	70.23	0.893
**2H**	1.39	359.4	3	4	4	1	70.23	2.310
**3H**	1.91	393.85	3	5	4	1	70.23	2.914
**4H**	1.53	377.39	3	5	4	1	70.23	2.452
**5H**	2.18	438.3	3	5	4	1	70.23	3.078
**6H**	−2.52	404.4	3	6	4	2	113.37	2.250
**7H**	1.14	389.43	3	5	4	2	79.46	2.152
**8H**	0.61	374.42	4	5	4	1	96.25	1.480
**9H**	1.66	402.47	3	5	4	2	73.47	2.416
**Etoposide**	1.27	588.56	3	13	2	5	160.83	1.15
**Etoposide ***	1.16	588.56	3	13	2	5	160.83	
**Etoposide #**		588.56	3	13	2			0.74

* https://www.drugbank.ca/drugs/DB00773. # *Chem. Pharm. Bull.*
**2013,**
*61*(12) 1228–1238.

**Table 9 ijms-21-01253-t009:** QSAR properties of the nine derivatives of TUDIHPP (1H-9H).

Comp.	Polarizability (*A^3^*)	Refractivity (*A^3^*)	Vol (*A^3^*)	Surface Area (Grid) *A^2^*	HE (*kcal/mol*)
1H	31.22	77.23	737.5	454.55	−10.87
2H	40.88	101.87	739.85	554.14	−11.73
3H	42.81	106.68	982.07	584.73	−11.43
4H	40.79	102.09	946.24	557.2	−11.47
5H	43.51	109.49	1000.77	590.22	−11.42
6H	42.72	108.18	998.3	586.26	−16.69
7H	43.35	108.33	1014.72	598.73	−13.42
8H	42.23	106.57	973.55	571.36	−15.84
9H	45.9	116.3	1070.91	621.57	−10.22
Etoposide	55.15	138.73	1448.08	808.26	−25.39
Etoposide *	58.77	139.02			
Etoposide #,+,†	55.5	140.1			

* https://www.drugbank.ca/drugs/DB00773; + https://smfmnewsroom.org/elements/etoposide-c29h32o13-structure/; # https://www.lookchem.com/Etoposide/. † https://www.lookchem.com/Etoposide/.

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
