# Peer review of "Density Functional Theory, Chemical Reactivity, Pharmacological Potential and Molecular Docking of Dihydrothiouracil-Indenopyridopyrimidines with Human-DNA Topoisomerase II"

_ijms, 2020, doi:10.3390/ijms21041253_

Round 1

Reviewer 1 Report

The authors have studied a series of dihydro thiouracil-4 indenopyridopyrimidines compounds regarding atomic and molecular properties together. Additionally molecular docking simulations were run against Topoisomerase II and their binding mode was compared to etoposide. Finally QSAR descriptors were predicted compared to etoposide.

In general the manuscript looks like a review article on methods selected and general rational drug design approaches. There are big paragraphs explaining the interaction on molecular docking while there should be clear figures with less details on manuscript.

The authors should discuss the known biological data for indeno pyrido pyrimidine UIPP derivatives and what they expect for the novel thiouracil analogues.

Three different approaches were chosen a) semi-empirical calculations (Ligand Based), b) docking simulations (Structure Based) and QSAR descriptors (Ligand Based). There is no liaison among them to produce a final predictive model.

Additionally many paragraphs describing known theories must be deleted. For example there is paragraph explaining LogP while only one phrase to describe the results. Same for QSAR studies, Lepinksi’s rules of 5 etc.

Most of the tables must be placed as supplementary data.

Headers on Table 6 must be aligned correctly.

Correct the A° with Å when referring to Angstroms.

The shape on Figure 7 must be corrected.

I think the manuscript must be very well re-written to provide a complete scientific story and give specific and clear results and conclusions.

Author Response

Reviewer 1 Comments:

Point 1: In general the manuscript looks like a review article on methods selected and general rational drug design approaches. There are big paragraphs explaining the interaction on molecular docking while there should be clear figures with less details on manuscript.

Response 1: Dully noted and done. Molecular docking simulations are summarized on manuscript with clear figures. The article becomes 25 pages.

Point 2: The authors should discuss the known biological data for indeno pyrido pyrimidine UIPP derivatives and what they expect for the novel thiouracil analogues.

Response 2: This is a novel class of molecules that has not been investigated before.  The proposed molecules can be potential anticancer drugs by comparison with etoposide which is a well- established anticancer drug and based on the proven outstanding biological activity of its analogous indeno pyrido pyrimidine UIPP derivatives [11]. The importance of the sulfur atom in drugs as sulfide or disulfide linkages provides great stability for the three dimensional structure within the molecule [10].

Point 3: Three different approaches were chosen a) semi-empirical calculations (Ligand Based), b) docking simulations (Structure Based) and QSAR descriptors (Ligand Based). There is no liaison among them to produce a final predictive model.

Response 3: The authors use computational medicinal chemistry to explore the novel dihydro thiouracil-based indenopyridopyrimidine (TUDHIPP) as potential anticancer drug. This manuscript is theoretical quantum chemical calculation with no laboratory experimental data. This is a novel class of molecules that has not been investigated before. The authors speculate on the advantages of single atomic replacements and molecular properties together. The authors use the computational technique Density Functional Theory (DFT) and molecular docking simulation to calculate properties including bond length, partial charges, dipole moment, HOMO. LUMO, binding energies, and QSAR to calculate properties including polarizability, volume and surface area. The authors discuss drug-likeness to establish pharmacological relevance. The authors use different approaches to study different point of view to get a final predictive results which help researches in next step of experimental approaches.

Point 4: Additionally many paragraphs describing known theories must be deleted. For example there is paragraph explaining LogP while only one phrase to describe the results. Same for QSAR studies, Lepinksi’s rules of 5 etc.

Response 4: Dully noted and done.

Point 5: Most of the tables must be placed as supplementary data.

Response 5: All data represent in tables are be used in discussion on manuscript.

Point 6: Headers on Table 6 must be aligned correctly.

Response 6: Dully noted and done.

Point 7: Correct the A° with Å when referring to Angstroms.

Response 7: Dully noted and done.

Point 8: The shape on Figure 7 must be corrected.

Response 8: Dully noted and done.

Point 9: I think the manuscript must be very well re-written to provide a complete scientific story and give specific and clear results and conclusions.

Response 9: The authors reformat more details on manuscript to provide a complete consistence scientific story and give specific and clear results and conclusions. 

Reviewer 2 Report

The authors use computational medicinal chemistry to explore the novel dihydro thiouracil-based indenopyridopyrimidine (TUDHIPP) as potential anticancer drug. This manuscript is theoretical using ab initio (HF, MP2, and B3LYP) with no laboratory experimental data. The authors speculate on the advantages of single atom replacements. The main structural elements are a para-substituted phenyl ring for a hydrogen and the para substituents include the halogens, amines, methyl ether and nitro. The authors use the computational technique Density Functional Theory (DFT) and molecular docking to calculate properties including bong length, partial charges, dipole moment, HOMO. LUMO, binding energies, and QSAR to calculate properties including polarizability, volume and surface area. The authors discuss drug-likeness and rely on Lipinski rules to establish pharmacological relevance, but this experienced reviewer knows that many successful new drugs do not obey Lipinski rules (i.e., Telaprevir, Venetoclax). The authors also use Veber’s rules, which are more applicable and Table 8 compares the Lipinski and Veber analysis; and Table 9 golden triangle with MW and Log D values.

Author Response

Response to Reviewer 2 Comments:

Point 1: English language and style are fine/minor spell check required.

Response 1: Dully noted and done.

Point 2: The authors use computational medicinal chemistry to explore the novel dihydro thiouracil-based indenopyridopyrimidine (TUDHIPP) as potential anticancer drug. This manuscript is theoretical using ab initio (HF, MP2, and B3LYP) with no laboratory experimental data. The authors speculate on the advantages of single atom replacements. The main structural elements are a para-substituted phenyl ring for a hydrogen and the para substituents include the halogens, amines, methyl ether and nitro. The authors use the computational technique Density Functional Theory (DFT) and molecular docking to calculate properties including bong length, partial charges, dipole moment, HOMO. LUMO, binding energies, and QSAR to calculate properties including polarizability, volume and surface area. The authors discuss drug-likeness and rely on Lipinski rules to establish pharmacological relevance, but this experienced reviewer knows that many successful new drugs do not obey Lipinski rules (i.e., Telaprevir, Venetoclax). The authors also use Veber’s rules, which are more applicable and Table 8 compares the Lipinski and Veber analysis; and Table 9 golden triangle with MW and Log D values.

Response 2: Thank you. This is a novel class of molecules that has not been investigated before.  The proposed molecules can be potential anticancer drugs by comparison with etoposide which is a well- established anticancer drug and based on the proven outstanding biological activity of its analogous indeno pyrido pyrimidine UIPP derivatives [11]. The authors use different approaches to study different point of view to get a final predictive results which help researches in next step of experimental approaches. The authors on manuscript provide a complete scientific story and give specific and clear results and conclusions.

Round 2

Reviewer 1 Report

The authors have made all corrections suggested. Thus the manuscript can be accepted in current form.